# G-Quadruplex Targeting in the Fight against Viruses: An Update

**DOI:** 10.3390/ijms222010984

**Published:** 2021-10-12

**Authors:** Emanuela Ruggiero, Irene Zanin, Marianna Terreri, Sara N. Richter

**Affiliations:** Department of Molecular Medicine, University of Padua, 35121 Padua, Italy; emanuela.ruggiero@unipd.it (E.R.); irene.zanin.1@studenti.unipd.it (I.Z.); marianna.terreri@studenti.unipd.it (M.T.)

**Keywords:** G-quadruplexes, virus, targeting, G-quadruplex ligands, antiviral therapy

## Abstract

G-quadruplexes (G4s) are noncanonical nucleic acid structures involved in the regulation of key cellular processes, such as transcription and replication. Since their discovery, G4s have been mainly investigated for their role in cancer and as targets in anticancer therapy. More recently, exploration of the presence and role of G4s in viral genomes has led to the discovery of G4-regulated key viral pathways. In this context, employment of selective G4 ligands has helped to understand the complexity of G4-mediated mechanisms in the viral life cycle, and highlighted the possibility to target viral G4s as an emerging antiviral approach. Research in this field is growing at a fast pace, providing increasing evidence of the antiviral activity of old and new G4 ligands. This review aims to provide a punctual update on the literature on G4 ligands exploited in virology. Different classes of G4 binders are described, with emphasis on possible antiviral applications in emerging diseases, such as the current COVID-19 pandemic. Strengths and weaknesses of G4 targeting in viruses are discussed.

## 1. Introduction

G-quadruplexes (G4s) are noncanonical arrangements that occur in guanine (G)-rich DNA and RNA strands during key cellular processes. They are characterized by the stacking of two or more G-tetrads, in which four Gs are linked through Hoogsteen-type H-bonds, centrally coordinated by monovalent cations (Figure 1A). G4s are highly polymorphic structures, the topology of which relies on several factors, such as strand stoichiometry, strand polarity, the anti or syn conformation of the glycosidic bond between the G base and the sugar, loop nature, length and location within the sequence, and the cellular environment. G4s can form intramolecularly within a single G-rich strand, or by the intermolecular polymerization of separate filaments, overall providing different three-dimensional conformations [1,2,3]. G4s have been extensively investigated for their role in cancer, since they are located in pivotal genomic regions, such as oncogene promoters, telomeres, transcription factor binding sites, and recombination sites [4]. G4 targeting has been widely exploited as an anticancer strategy, leading to the development and optimization of several G4 ligands, two of which have entered clinical trials [5]. Research on G4s in cancer has unraveled many biological pathways, providing new insights on the role of G4s in cancer pathogenesis, and corroborating the possibility to target these structures for therapy [4]. This evidence prompted investigation of G4s also in different organisms, such as yeasts [6], bacteria [7,8], and viruses [9], where G4s emerged as important elements in the regulation of these microorganisms’ biological processes.

Viruses are entities that are billions of years old, and their existence traces back to the first cellular organisms [10]. Viruses’ necessity to exploit host macromolecule synthesis machineries for their own replication makes them powerful intruders: throughout evolution, viruses have parasitized all living beings, from bacteria and archaea to animals, plants, and humans, influencing species evolution [10]. From a structural point of view, the viral particle, called a *virion*, consists of a protein shell, the *capsid*, which encloses the virus nucleic acids, either DNA or RNA in a single- or double-stranded form. The virion may possess an additional external lipid membrane, the *envelope* (Figure 1B). The Baltimore classification allocates viruses into seven groups, according to the nucleic acid type and the mRNA synthesis pathway (Figure 1C). To replicate, viruses exploit host cell substrates and pathways, which provide the essential elements required to assemble and release new virions. In some cases, viruses do not actively replicate within the host cell, but rather establish a latent infection in which the virus evades the host immune system and often lasts for the patient’s life [11]. Viral infections span from being asymptomatic to causing severe, persistent, or even deadly diseases. They may spread abruptly within the human population, leading to epidemics that affect health, national economies, and the overall well-being of societies. The current COVID-19 pandemic, caused by the SARS coronavirus 2 (SARS-CoV-2) and ongoing since December 2019, is the dramatic memento of the impact of viral infections, which keep challenging humanity [12].

In the past years, technological advances successfully provided innovative antiviral therapeutics, such as the pan-genotypic direct-acting antiviral drugs against the hepatitis C virus (HCV), and the anti-retroviral therapy employed to control the human immunodeficiency virus (HIV). Nonetheless, most available therapies fail to accomplish virus eradication or sustained virological response [13], and no drugs exist against the majority of human viruses, making identification of new antiviral targets, and the consequent development of innovative antiviral agents, of the utmost importance.

Viruses belonging to almost all Baltimore classes have been found to be enriched in highly conserved and putative G4-forming sequences (PQSs) located in significant position within the viral genome [14,15]. The development and optimization of bioinformatics tools to detect G4 motifs have highly accelerated the identification of viral PQSs; interestingly, most of the identified sequences were proved to actually fold into G4s in vitro [16]. In this context, employment of known G4 ligands provided an excellent tool to unravel G4-mediated viral pathways, eventually proving that G4 targeting can be a promising and innovative approach in antiviral therapy [17].

To assess the importance of G4s in virology, several reviews that report findings in the field from different points of view, such as the G4 itself [16,18,19] or the pathogen of interest [9,20,21], have been very recently published. In this review, we focus on the G4 ligands employed as antiviral agents, aiming to provide a comprehensive update covering the available literature, with major focus on the last three years. Only human viruses have been considered, with a final analysis that includes the recent advances against the SARS-CoV-2 virus.

## 2. G4 Ligands

To date, several compounds have been developed to target G4 structures, leading to the establishment of new promising therapeutic approaches not only against cancer, but also in the antiviral field (see [22] for a comprehensive list of verified G4 ligands). Generally, G4 binders are characterized by a polycyclic planar aromatic scaffold with protonable side chains: interaction with G4 structures generally occurs via π-π stacking with the terminal G-tetrads, while lateral positive charged moieties interact with the phosphate groups in loops and grooves [23]. In the last decade, great effort has been made to improve binding and selectivity of specific ligands towards G4s, leading to several chemical classes with promising G4 binding activities. Most G4 ligands in cancer cells decrease cell growth due to changes in telomere maintenance, oncogene expression, and genome stability [4,5]. A few G4 ligands have also entered the early phases of clinical trials, corroborating the feasibility of a G4-targeted therapy, even though none has progressed to phase II, principally due to bioavailability issues [5]. Since most of the bioinformatically identified PQSs within viral genomes present high conservation rates, especially of the Gs involved in G4 formation [14], a G4-based antiviral therapy could potentially be effective against different viral strains, circumventing the frequent viral recombination events. Thus, application of G4 ligands in the antiviral field may pave the way for innovative strategies in the fight against viruses.

The following sections describe the G4 ligands so far reported to exert antiviral activity: for each compound, a general overview on its structural features and G4 binding efficiency, with focus on its antiviral properties, are provided.

### 2.1. TMPyP4

Compound 5,10,15,20-tetrakis-(*N*-methyl-4-pyridyl)porphyrin (TMPyP4, Figure 2) is one of the most extensively studied G4-binding molecules, because its physicochemical features, such as size, planar core, positively charged substituents, and hydrophobicity, make it fit to stack on the G4 scaffold. Moreover, the existence of TMPyP2 (Figure 2), a structural isomer with the *N*-methyl groups in the sterically hindered 2-position that preclude its binding to the G4 and thus makes it the perfect negative control molecule, triggered the employment of porphyrins in G4 research [24]. TMPyP4 was proved to induce telomerase inhibition upon binding to telomeric DNA G4s in cancer cells [25,26,27,28,29,30], and to downregulate expression of oncogenes, such as *c-myc* [31,32], *k-ras* [33], and *bcl2* [34] upon interaction with the G4 region located in their protomer sequences [35]. TMPyP4 binds all types of G4 conformations, but also different noncanonical DNA structures as i-motifs, intercalated arrangements that can be formed in cytosine-rich strands opposite to G4s [36,37,38,39,40]. Its main weakness, however, is its mild selectivity for quadruplex over duplex DNA [41]. TMPyP4 generally increases G4 stability independently from the nature of the G4-forming sequences by stacking its aromatic core on the upper or lower G4 tetrad and engaging in additional noncovalent bonds [24,38,42,43]. However, interaction with RNA G4s also has been associated with a destabilizing effect [44,45,46,47].

Due to its low cytotoxic profile, TMPyP4 has been used in viruses to evaluate the effect of G4 stabilization as therapeutic strategy: data on several human herpesviruses (HHVs), human papillomavirus (HPV), and Zika virus (ZIKV) have been reported (Figure 3).

HHVs are massively widespread within the human population, causing life-long infections that lead to the onset of a series of diseases such as orolabial and genital herpes, mononucleosis, and cancer. Herpes simplex viruses 1 and 2 (HSV-1, HSV-2) and Varicella zoster virus (VZV) belong to the *alpha* subfamily, the most studied group in G4 research; human cytomegalovirus (HCMV), HHV-6, and HHV-7 are grouped in the *beta* subfamily; and the Epstein-Barr virus (EBV) and the Kaposi’s sarcoma-associated virus (KSHV), two viruses that can induce tumorigenesis in humans, are classified as *gamma*-HHVs. Herpesviruses are all characterized by a large dsDNA genome, remarkably enriched in GC content and PQSs [48].

In HSV-1, the compound was tested in infected cells, where it interacted with and stabilized the most abundant HSV-1 G4-forming sequences present in the repeated regions of the viral genome. The compound did not influence HSV-1 entry nor replication, but it induced trapping of fully infectious HSV-1 virions in vesicles, independently of autophagy [49]. In HCMV, TMPyP4 was evaluated on 36 PQSs associated with 20 viral genes: the porphyrin increased G4s’ stability without affecting their conformation [50]. In both KSHV and HCMV viruses, G4-forming motifs were identified upstream of the microRNAs (miRNAs) cluster (miR-K12-1-9,11) and miR-US33 promoters, respectively. Viral miRNAs are critical for viral latency maintenance due to their inhibition of the expression of viral modulators and host genes [51,52]. Treatment with TMPyP4 destabilized miRNA G4s: ΔT_m_ of −20 °C for KSHV miRNA G4 and −15 °C for HCMV miRNA G4, with respect to untreated conditions, were observed in CD analysis [53]. However, this effect resulted in the opposite outcome: enhancement of KSHV miR-K12 cluster promoter activity and inhibition of HCMV miR-US33 promoter activity. This suggests that G4s in the regulatory region of HHVs-encoded miRNAs may play a different role in infection modulation. In KSHV, TMPyP4 additionally stabilized G4 structures identified in the mRNA of the latency-associated nuclear antigen (LANA) [54], which is the protein most expressed during latency and fundamental for viral transmission and host immune surveillance evasion [55,56,57]. After treatment with the compound for 24 h, Dabral et al. observed reduction of LANA levels in KSHV-infected cells, supporting the hypothesis that G4 stabilization in LANA mRNA leads to translation inhibition: stabilization of LANA mRNA G4s resulted in lower surface expression and LANA antigen presentation in KSHV-positive cells, and supported KSHV infection maintenance [54].

HPV is the DNA virus responsible for the hyperplastic lesions in epithelial structures that, in the case of high-risk strains, evolve in cervical and oropharyngeal cancers. Several high-risk HPV types have been analyzed for the presence of PQSs [58]. Incubation of TMPyP4 with PQSs identified in seven different HPV strains; i.e., HPV9, HPV16, HPV18, HPV32, HPV52, HPV57, and HPV58, reported high G4 stabilization in all the tested sequences, as assessed by FRET and CD melting experiments [59].

Zika virus (ZIKV), an arthropod-born virus, caused a major outbreak in 2016 that evolved into a public-health emergency, since infection may induce severe neurological complications in infants and adults. The virus presents a positive-sense RNA genome that contains several PQSs distributed in genes and untranslated regions [60]. TMPyP4 was reported as a stabilizing agent of G4 structures identified in ZIKV. In vivo experiments revealed that the compound inhibits viral growth, as well as viral genome replication and protein expression, in a dose-dependent manner, indicating a new possible strategy against ZIKV infection [61].

TMPyP4 is a potent compound to be used in antiviral assessment; however, because of its low G4 versus duplex DNA selectivity [38], appropriate controls must be included to assess if G4-independent mechanisms of action are also present.

### 2.2. PhenDC3 and Bisquinolinium Derivatives

PhenDC3 (Figure 2) is the most representative member of the bisquinolinium family, which interacts with G4s through extensive π-stacking with the external G-tetrads [62]. It exhibits high G4 affinity that results in high selectivity for nuclear G4 DNA over the predominant duplex DNA. The peculiar characteristic of this compound is its ability to adopt an internally organized H-bonded *syn*-*syn* conformation, which is critical for G4 binding [63]. Initially studied as an anticancer agent due to its ability to affect telomerase processivity [63], it is also now employed in the viral field, with reports on HCV, HPV, and EBV viruses (Figure 4).

HCV is one of the major causes of liver cancer. It has a negative-sense RNA genome organized in a unique open reading frame that produces structural and nonstructural viral proteins. Several PQSs have been characterized within the HCV genome, with the most stable G4 located in the core gene [43]. In HCV, PhenDC3 was able to bind and stabilize a highly conserved G4-prone sequence (HCV110-131) located in the stem-loop IIy’ at the 3′-end of the HCV genome. The compound stabilized the target G4 by 7.5 °C in melting experiments. Jaubert et al. showed that PhenDC3 hampered in vitro RNA-dependent RNA synthesis by HCV polymerase and inhibited viral replication in cells, in conditions where cell viability was not affected [64].

PhenDC3 has been reported as a strong in vitro stabilizer of most HPV G4s by CD, FRET, and fluorescent intercalator displacement assay (TO displacement). However, notwithstanding the promising in vitro results, the compound failed to reduce viral replication and protein expression in organotypic raft cultures [59].

Most PhenDC3 studies in virology were performed on EBV. The mRNA of its key viral protein EBNA1 is enriched in G4s that are recognized by the cellular protein nucleolin (NCL), the binding of which results in downregulation of EBNA1 expression. PhenDC3 was found to displace NCL from EBNA1 mRNA, with consequent increase of EBNA1 translation and activation of antigenic peptide production [65]. Due to its affinity towards G4s and the observed effects against EBV, PhenDC3 was used as a prototype for the design of new putative G4 ligands. Reznichenko et al. recently developed 20 cationic bis(acylhydrazone) derivatives that were tested for their in vitro and in vivo G4 binding activity in EBV-related models. Among these molecules, PyDH2 and PhenDH2 (Figure 2) were identified as the most promising compounds due to their ability to bind EBNA1 mRNA G4s, enhancing EBNA1 expression in a glycine-alanine repeat (GAr)-dependent manner and preventing EBNA1 G4/NCL interaction, and overall hampering EBV immune evasion. Moreover, PyDH2 and PhenDH2 were reported to be significantly less toxic than the model drug PhenDC3, therefore representing promising players in G4-mediated antiviral therapy [66].

### 2.3. NMM

N-methyl mesoporphyrin IX (NMM, Figure 2) is an N-core methylated nonplanar derivative of mesoporphyrin IX. Initially identified as an inhibitor of Fe^2+^ insertion into protoporphyrin IX by ferrochelatase, the terminal enzyme in heme biosynthetic pathway [67], NMM was one of the first small molecules reported to bind G4s. It displays unique selectivity not only for G4 versus duplex DNA, but also for parallel versus antiparallel G4s [68]. Indeed, the N-methyl group of NMM fits into the center of the parallel G4 core, where it aligns with potassium ions, giving rise to an efficient π-π stacking that confers a strict balance to the G-tetrad [69]. NMM has the peculiarity of being a fluorescent molecule (excitation at λ = 393 nm and emission at λ = 610 nm) that, when selectively bound to G4s, considerably amplifies its signal (up to 60-fold in the case of parallel G4 binding). Due to its preferential binding to a specific G4 topology, NMM has the potential to discriminate between different strand orientations, based on its fluorescence fold enhancement [70]. This feature allowed the development of a simple, highly sensitive, and selective method used in G4 investigation [71,72,73,74]. In cells, NMM proved to regulate gene expression, influence the epigenetic landscape, and affect cell growth through G4-mediated mechanisms [75].

NMM applications in the antiviral field are summarized in Figure 5. NMM has been recently used in the analysis of 36 HCMV PQSs, which were associated with 20 genes. Biophysical analysis showed that most G4s in the HCMV gene regulatory regions actually folded into stable G4s, but only a few of these suppressed gene expression when examined in in vivo assays. In in-cell reporter assays and HCMV-infected cells experiments, Ravichandran et al. pinpointed that gene suppression was specifically observed when G4s were located within gene promoters. In this context, NMM exhibited general stabilization of parallel G4 structures, while no effect on a unique HCMV antiparallel G4 was observed, thus confirming NMM selective binding to parallel G4s [50].

NMM was employed in biophysical analysis of RNA PQSs recently identified in the influenza A virus (IAV) genome. IAV is responsible for most influenza outbreaks, since it can affect fowl, humans, and other mammals. Its negative-sense RNA genome is structured in eight segments that encode for viral proteins. Despite the well-known high variability of this virus, which is IAV’s key survival mechanism, several conserved G4 motifs were identified. NMR and native polyacrylamide gel electrophoresis analysis upon incubation with NMM corroborated the parallel G4 folding of the tested sequences [76].

### 2.4. Naphthalene Diimides

Naphthalene diimides (NDIs) are among the most versatile G4 binders, and are widely employed in biomedical applications. Chemically, they present a large, flat aromatic core that interacts by π-π stacking with the G-quartet; they are functionalized with up to four protonable side chains, which in turn allow formation of H-bonds with the phosphate groups in the G4 grooves. The resulting G4-NDI complex is extremely stable, making NDIs excellent tools for diagnostic and therapeutic investigations. A thorough medicinal chemistry approach on the NDI scaffold led to the development of several derivatives with increased affinity and selectivity towards G4s: these compounds were mainly employed in anticancer research because of their ability to tightly bind G4s located at telomeres and oncogene promoters [77]. Among NDI derivatives, core extended NDIs (c-exNDIs) showed exceptional solubility in water [78]; they were initially tested against HIV-1 and HSV-1, showing remarkable antiviral effects, with higher affinity for viral over cellular G4s, and were the first compounds to provide solid evidence of the possibility to exploit G4s in antiviral approaches [78,79].

Recently, the core extended c-exNDI 2 (Figure 2) was employed in the analysis of PQSs identified in the long terminal repeat (LTR) region of the integrated genome of all RVs (Figure 5) [80]. RVs, the most representative of which is the HIV-1 virus, present the unique characteristic of integrating their positive-sense RNA genome into the host DNA upon retrotranscription: as a consequence, the virus establishes latency and cannot be eradicated from the host, causing deadly diseases [9]. C-exNDI 2 showed stabilization of all tested sequences in CD and *Taq* polymerase stop assays: the latter revealed that G4 stabilization deeply impacted on polymerase progression [80]. NDIs were also conjugated to a set of peptide nucleic acid (PNA) sequences, complementary to the DNA sequence of interest to enhance specificity in G4 recognition. Recently, this new strategy was employed to target mutually exclusive HIV-1 LTR G4s, taking advantage of the cooperative interaction of the NDI moiety with the G4 and hybridization of the PNA with the targeted G4 flanking region. The conjugates were able to induce and stabilize the least-populated HIV-1 G4, to the detriment of the more stable one: this approach showed that control of the G4 landscape is possible, therefore representing a possible effective tool for wider applications [81].

### 2.5. BRACO-19 and Acridine Derivatives

The *N,N*′-(9-((4-(dimethylamino)phenyl)amino)acridine-3,6-diyl)bis(3-(pyrrolidin-1-yl)propanamide), known as BRACO-19 (B19), is a 3,6,9-trisubstituted acridine widely employed in G4 research due to its high selectivity for G4s versus duplex DNA. Among other tested regioisomeric compounds, the 3,6,9-trisubstituted acridine scaffold showed higher selectivity, lower cytotoxicity, and strong telomerase inhibition mediated by G4 recognition [82]. The basis of this interaction resides on its central planar pharmacophore, responsible for the binding to G-quartets through π- π interactions, while the two side chains with a tertiary amine moiety enable recognition of G4 grooves. As a matter of fact, protonation of these amino groups at physiological pH profoundly influences G4 interaction [82].

Application of B19 in G4 research has been widely extended to virology, due to its low cytotoxicity and high G4 binding activity; for these reasons, this compound is frequently used as first tool to investigate the presence and role of G4s in viral infections. Indeed, B19 has been used in DNA viruses, such as different HHVs, human adenovirus (HAdV), HPV, and parvovirus; and in RNA viruses, such as RVs and ZIKV (Figure 6).

In *alpha*-HHVs, B19 was employed to characterize PQSs located in the immediate early (IE) gene promoters. Biophysical and biochemical analysis demonstrated the actual folding of selected sequences into stable G4s. Subsequent CD thermal unfolding analysis indicated a large increase in T_m_ upon incubation with B19, confirming the stabilizing effect exerted by this ligand. To assess the biological role of G4s in *alpha*-HHV IE gene transcription, the luciferase firefly gene was cloned downstream the promotor sequence of two representatives IE genes from HSV-1. Treatment with B19 affected the activity of both promoters in a dose-dependent manner: the final outcome of this stabilization was the consistent impairment of IE gene transcription [83]. In this line of research, the viral transcription factor ICP4 was proved to interact and unfold G4 structures in HSV-1 IE gene promoters, including those located in its own promoter, promoting viral transcription. A luciferase assay with the ICP4 promoter in the presence of B19 resulted in a reduced promoter activity, ascribable to the competition between B19 and ICP4 for the promoter region. Since ICP4 is essential for the viral replication cycle, this type of strategy that targets ICP4-G4 interaction could represent a novel therapeutic approach [84]. B19 had been previously shown to also bind to G4s located in noncoding regions of the HSV-1 genome, with a decrease in viral DNA and viral late protein synthesis, corroborating a crucial role for G4s in the regulation of *alpha*-HHVs’ life cycle [85].

B19 was used to characterize the presence of G4s in the HAdV genome. HAdVs are nonenveloped dsDNA viruses that can cause severe ophthalmic, respiratory, and neurological diseases. HAdV genomewide bioinformatics investigation revealed several conserved PQSs, the folding of which was proved through biophysics studies. Treatment with B19 highly improved HAdV G4 stabilization; in addition, using a recombinant AdV plasmid containing the green fluorescent protein (eGFP) reporter gene, B19 showed decreased fluorescence emission and inhibited virion production in HEK293 cells [86].

In HPV, application of B19 on viral G4s was performed along with C8, an acridine derivative. Both compounds were found to bind with high affinity by fluorescent intercalator displacement assay (G4-FID). C8 was also evaluated for its antiviral activity in organotypic epithelial cultures infected with HPV16 or HPV18: C8 treatment for 20 days was successful in reducing viral replication, in contrast to PhenDC3, which was inactive in vivo. Since there is no treatment able to eradicate HPV due to the establishment of latent reservoirs, an approach that targets G4s in the viral genome could address both the replicating and latent virus [59].

Among DNA viruses, compound B19 was also tested against parvovirus-B19, the pathogen responsible for the onset of fifth disease in children. Two PQSs located in proximity of the origin of replication were identified; however, despite bioinformatic prediction, neither biophysical and biochemical analysis, nor treatment with the drug supported G4 folding. Furthermore, the G4 ligand B19 did not display any antiviral activity on EPCs cells infected with the parvovirus-B19V strain, demonstrating that bioinformatic prediction of PQSs does not always reflect the actual existence of G4s [87].

Among RNA viruses, B19 activity was tested on G4s localized in the RVs LTR U3 region. CD spectroscopy demonstrated that several sequences, retrieved from all RVs subfamilies, formed stable G4 structures in the presence of B19. Addition of the ligand also decreased full-length products in *Taq*-polymerase stop assay due to the stalling of polymerase at the stabilized G4s [80]. Among RVs, G4s were extensively studied for their role in HIV-1 regulation. B19 was proved to exert an antiviral effect against HIV-1, affecting both pre- and postintegration steps during the viral cycle. This behavior could be explained by the presence of G4s in the U3 region of both the RNA genome and the integrated proviral DNA. In fact, B19 efficiently inhibited reverse transcriptase progression, preventing DNA synthesis during the preintegration step, and polymerase on the DNA template, thus inhibiting viral transcription and replication at the postintegration step [88]. HIV-1 G4s were also found to be bound by the viral nucleocapsid protein NCp7, which exerts G4 unfolding activity. An electrophoretic mobility shift assay and CD analysis revealed that NCp7 binds RNA G4s, promoting formation of the DNA/RNA intermediate that forms during reverse transcription. B19 was able to partially counteract NCp7 G4 unfolding activity. Therefore, G4-mediated inhibition of HIV-1 at the RNA level exerted by B19 is due to both RT stalling and NCp7 chaperone activity inhibition [89].

Finally, in Vero cells infected with MR766 ZIKV, B19 was able to reduce viral protein expression as well as genome replication, with an overall antiviral effect likely mediated by stabilization of PQSs identified at the viral RNA level [61].

### 2.6. Pyridostatin and Derivatives

Pyridostatin (PDS) is a G4-binding small molecule presenting all the chemical features of an effective G4 binder: an electron-rich, flat, aromatic surface, and the ability to participate in hydrogen bonding. PDS stabilized telomeric G4s, leading to the inhibition of telomere elongation and triggering DNA-damage response to the telomeres. From the PDS scaffold, many derivative compounds were developed and proved to be effective in inhibiting cancer cell line replication through a G4-mediated mechanism of action [90].

PDS and its analogues have been widely used against several HHVs, HPV, HCV, and very recently, SARS-CoV-2 (Figure 7).

In KSHV and HCMV, PDS was employed, analogously to TMPyP4, to study G4-forming sequences located in the promoter proximal to miRNAs (miR-K12 in KSHV, miR-US33 in HCMV). CD spectroscopy revealed that both sequences formed stable parallel G4s further stabilized by PDS. G4s’ biological role was investigated through a firefly luciferase assay: HEK293T cells were transfected with the G4 region from both viruses upstream of the firefly luciferase gene and treated with increasing concentration of PDS, which exerted opposite effects on the two promoters, as also observed for TMPyP4. It was noted, however, that in contrast to PDS, TMPyP4 destabilized G4s, and therefore the final outcome could be ascribed to multiple pathways, possibly unrelated to G4s [53].

In EBV, PDS was used to investigate NCL binding to the G4s located in the GAr domain of EBNA1 mRNA. PDS was suggested to hinder protein-mRNA binding, thus triggering the immune response. However, PDS was not able to reduce the interaction between NCL and EBNA1 mRNA in a proximity ligation assay. In addition, PDS did not inhibit NCL pulldown with beads coated with EBNA1 G4: weaker affinity of PDS for this specific G4 structure or alteration of PDS/NCL/ENBA1 complex was suggested to explain these results [65].

PDS bound to PQSs from seven different HPV genotypes, but with lower affinity with respect to other tested G4 ligands; in vivo investigation is still needed to ascertain PDS anti-HPV activity [59].

In HCV, a PDS analogue known as PDP was employed to investigate NCL/viral RNA G4 interaction. PDP competed for G4 binding as shown by the decrease of pulled-down NCL. This result was further confirmed by immunofluorescence analysis performed in Huh-7.5.1 cells transfected with the HCV genome: colocalization of NCL and the G4-specific antibody BG4 was reduced upon incubation with PDP [91].

Very recently, the antiviral effect of PDS in ZIKV infection was reported. Upon confirmation of PDS binding and stabilization of three G-rich ZIKV RNA sequences, its ability to recognize RNA G4s in cell cytoplasm was tested. Vero cells were transfected with the most stable RNA G4: addition of PDS resulted in high increase of RNA G4 signal, visualized by immunofluorescence analysis. Next, the antiviral effect was analysed using Vero cells infected with ZIKV. Treatment with PDS reduced both the ZIKV cytopathic effect and mRNA synthesis, with stronger inhibition during postinfection treatment. PDS was also found to affect ZIKV NS2B-NS3 protease activity, which is essential for viral protein production. Taken together, these results support the potential role of PDS as an anti-ZIKV agent thanks to multitarget inhibitory effects [92].

### 2.7. Other Ligands

Further G4 ligands tested as antiviral agents included compound CX-5461, discovered in a molecular screening to find selective inhibitors of Pol I transcription, conducted by Drygin et al. [93]. The compound was initially found to induce senescence and autophagy, ultimately leading to cell death in cancer cells with respect to normal ones. Studies conducted in murine xenograft models revealed good pharmacokinetics and an antitumor effect [93,94]. It was later demonstrated that CX-5461 anticancer activity was G4-mediated [95], and currently, CX-5461 is under clinical trials for the treatment of solid tumors. More recently, the compound was tested against HCMV: it successfully inhibited virus replication and pre-RNA synthesis in human fibroblasts infected with HCMVTB40/E strain (Figure 5). Surprisingly, treatment with CX-5461 48 h postinfection resulted in greater reduction of viral replication, with disruption of viral DNA synthesis and late protein production. This results likely stemmed from the compound activity on the cell, which in turn was less able to sustain viral infection, or alternatively by activation of the cell immune response; indeed, it has been reported that the compound activated a cellular stress response that contributed to inhibition of viral growth [96].

Finally, a series of benzoselenoxanthene derivatives were developed and tested against IAV virus (Figure 5). TMPRSS2, a transmembrane serine protease able to cleave IAV hemagglutinin, is crucial for virus entry into the host cell: TMPRSS2 promoter contains a G-rich sequence capable of folding into G4, the stabilization of which could inhibit TMPRSS2 expression. Seven benzoselenoxanthene derivatives were prepared with the purpose of increasing G4 recognition: among them, four displayed stabilizing effect in in vitro experiments. In addition, benzoselenoxanthenes efficiently decreased TMPRSS2 expression in Calu3 cells, as well as viral replication, in a dose-dependent manner [97]. Since TMPRSS2 is involved also in SARS-CoV-2 entry, these novel compounds could represent a novel antiviral strategy against both IAV and SARS-CoV-2. Indeed, in a very recent work, the anti-SARS-CoV-2 activity exerted by ribavirin, a well-known antiviral agent, was related to G4 modulation at the TMPRSS2 promoter [98]. Additional experiments will need to be performed to confirm this fascinating hypothesis.

## 3. Application of G4 Ligands in Emerging Viruses: The Case of SARS-CoV-2

The ongoing SARS-CoV-2 pandemic has been one of the most challenging health crisis faced by modern society. The urge of feasible therapies has boosted researchers worldwide to search for new strategies to fight the virus: one of these is the use of G4s as novel antiviral targets. Several bioinformatics tools were employed to assess the presence of PQSs within the SARS-CoV-2 RNA genome, showing very low enrichment in G4s [8,99,100,101]. Nonetheless, the identified PQSs were shown to fold into two-layered G4s and to be located within the open reading frames (ORFs) 1ab and 3a, spike, membrane, and nucleocapsid genes, suggesting that RNA G4s could be involved in the regulation of viral replication, assembly, and immune-response modulation [99]. SARS-CoV-2 G4s were stabilized by TMPyP4 and B19 ligands in a primer extension assay, in which treatment with G4 ligands resulted in an overall reduction of PCR product yield. In addition, in constructs including the G4 regions upstream the GFP gene, G4 ligands treatment reduced GFP expression [102]. SARS-CoV-2 RNA G4s were found to be targeted and unwound by the cellular protein CNBP, disclosing new aspects in the virus/host interplay: since CNBP expression is enhanced in response to viral infections, the protein/G4 interaction could be targeted for antiviral purposes [101]. The SARS-Unique Domain (SUD) of SARS-CoV Nsp3 protein had been previously shown to interact with G4s, and this interaction was proposed to be crucial for SARS genome transcription/replication [103]. Primary structure alignment of SARS-CoV and SARS-CoV-2 SUD proteins revealed high similarity of the corresponding macrodomains, coupled with high conservation of the amino acids necessary for G4 binding and viral replication. Thus, disruption of the SUD/G4 interaction could constitute an alternative antiviral approach [104]. To this end, some G4 ligands were recently employed in the in vitro investigation of SARS-CoV-2 SUD interaction with G4-forming sequences. Different compounds were tested: the bisquinolinium derivatives PhenDC3, PhenDH2, and PDC (a PDS analogue); phenanthroline derivatives; and metallated porphyrins. With a Homogeneous Time-Resolved Fluorescence (HTRF) assay using a cellular RNA G4 sequence (TRF2), Lavigne et al. showed that all tested molecules inhibited SUD/G4 interaction, with IC_50_ values between 15 and 50 nM. The three most stable PQSs predicted in SARS-CoV-2 genome were also tested, but none of them interacted with the SUD domain, suggesting a preferential interaction with host cell DNA or RNA, rather than viral RNA [104].

Additional G4 analysis in the SARS-CoV-2 genome also located a G-rich sequence in the viral nucleocapsid phosphoprotein (N) gene, which was shown to fold into RNA G4 and be further stabilized by the addition of the PDP ligand. G4 formation was monitored in live cells using EGFP as reporter gene: upon treatment with PDP, a large decrease in fluorescence emissions was observed by both flow cytometry and immunofluorescence analysis. Furthermore, PDP inhibited N mRNA translation, which resulted in decreased protein expression: since the N protein is involved in virus assembly and replication; its inhibition mediated by G4 recognition could represent a novel therapeutic approach in the fight in the COVID-19 pandemic [105].

## 4. Discussion and Future Developments

Recent research on viral G4s and their targeting with G4 ligands has validated this approach as a possible new antiviral strategy. This is supported by the fact that the identified viral PQSs are generally located in highly conserved regions. Base-conservation analysis is indeed a critical virological parameter, since viral mutation rates are normally very high and also pose a limit to current antiviral therapies [106]. The poor availability of deposited sequences for many viruses, especially the emerging ones, hinders an exhaustive conservation analysis; however, evidence collected so far unquestionably points to G4s as conserved elements within viral genomes, thus supporting their exploitation as antiviral targets.

The most demanding task in the development of antiviral G4 binders is conferring them the ability to discriminate between viral and cellular G4s. As described above, current G4 ligands share common chemical features that make them selective for G4s over duplex DNA through stacking interactions with the G-tetrad. This property, combined with high molecular weight and positively charged substituents, reduces the selectivity towards different G4s and confers poor druggability to the ligands, usually preventing them to progress to clinical trials. Ad hoc design of novel molecules based on deep structural resolution of viral G4s could enhance selectivity: since G4s differ mainly in the loop and groove regions, structural characterization of these G4 moieties would provide the basis for the design of more selective ligands. Unfortunately, research in this direction is still poor.

Notwithstanding their scarce selectivity, several G4 ligands showed promising G4-dependent antiviral activity. To make sense of this outcome, both the virus and the host must be considered. During infection, the virus extensively replicates its genome to create a large number of new virions, therefore the amount of viral G4s considerably exceeds that of cellular G4s [107]. The antiviral effect observed upon G4 ligand treatment is likely due to the overall G4 stabilization that impairs key viral pathways. However, different G4 ligands were reported to induce different effects on the same viral G4 substrate: PhenDC3 stimulated EBV EBNA1 mRNA synthesis, while PDS led to the exact opposite result [108,109]. In addition, while most studies on viruses corroborate a repressive role for G4s in viruses, stabilization of the G4 located in the hepatitis B virus *preS2/S* gene resulted in enhanced viral gene expression. Therefore, deeper characterization of G4-mediated viral mechanism at the molecular level is needed to disentangle this complex picture. The second key aspect is the host response to the viral infection. Most of the identified protein/viral G4 interactions involve a cellular G4-binding protein. NCL, for instance, through binding to viral G4s, silences gene and viral transcription in HCV and HIV-1 [91,110], eventually reducing virus propagation. The protein also stabilizes EBV EBNA1 G4s, downregulating EBNA1 expression and antigen presentation, and thus limiting viral immune evasion [109]. It follows that the final G4-mediated antiviral effect exerted by G4 ligands needs to be ascribed to multiple mechanisms of the virus–host interplay. Although most of these pathways still need to be elucidated, they could potentially represent ultimate targets for successful G4-based antiviral therapy.

One of the most thrilling exploitations of antiviral G4 ligands is the possibility to target latent viruses, i.e., viruses that stay in the host in a dormant state and that could be reactivated when conditions are favourable. Eradication of a latent virus from its host has not been achieved yet, and consequently, infective agents such as the entire range of HHVs, HPV, and HIV-1 remain with the infected human host for the host’s entire life, and antiviral therapy must be maintained to avoid disease resurgence upon virus reactivation. A G4-based targeting strategy would influence both the replicating and the latent virus, thus possibly leading to virus eradication. In this context, both viral and host chromatin manipulation need to be considered. Indeed, epigenetic modifications of the viral genome could trigger either a lytic or latent infection [111]. On the other hand, the host chromatin status, active or repressive, can determine the final outcome of the viral infection [112]. Increasing evidence supports a possible G4 involvement in both cases [19,113,114]; however, the mechanisms underneath still need to be elucidated. Therefore, further investigation of the role of G4s will help unravel fundamental aspects of virus-host interaction.

In conclusion, all data on G4s in viruses point to G4 ligands as a possible new successful approach in the management of viral infections, a strategy that could potentially lead to the development of cutting-edge therapeutic approaches in the treatment of fatal human diseases related to viral latency, such as AIDS and virus-induced cancers.

## Figures and Tables

**Figure 1 ijms-22-10984-f001:**
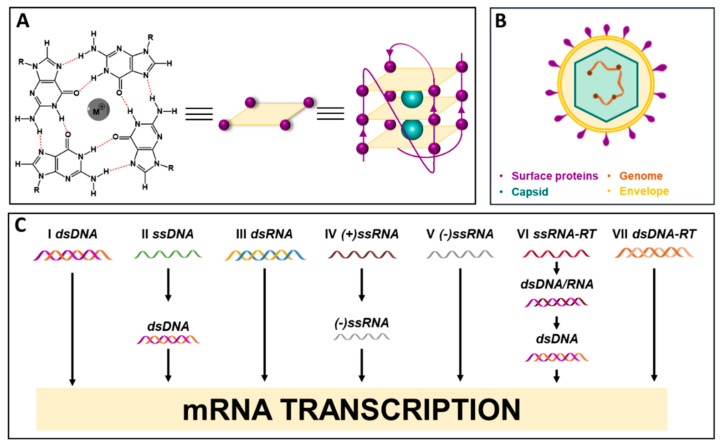
(**A**) Chemical structure of G-tetrad and G-quadruplex assembly; (**B**) virus structure and main components; (**C**) Baltimore classification of viruses.

**Figure 2 ijms-22-10984-f002:**
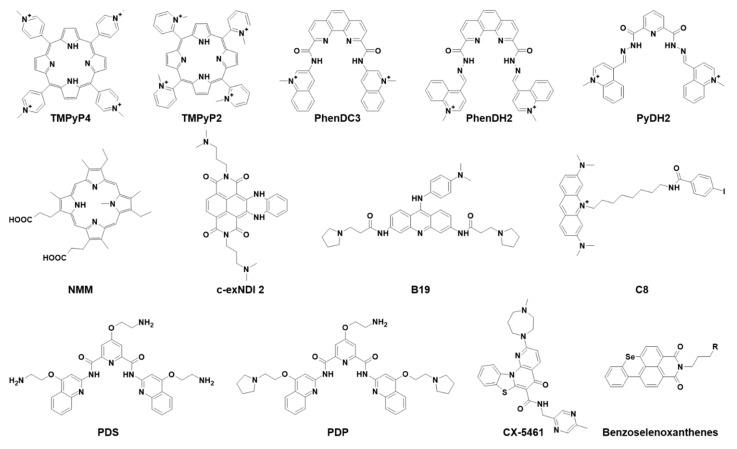
Chemical structures of G4 ligands investigated as antiviral agents.

**Figure 3 ijms-22-10984-f003:**
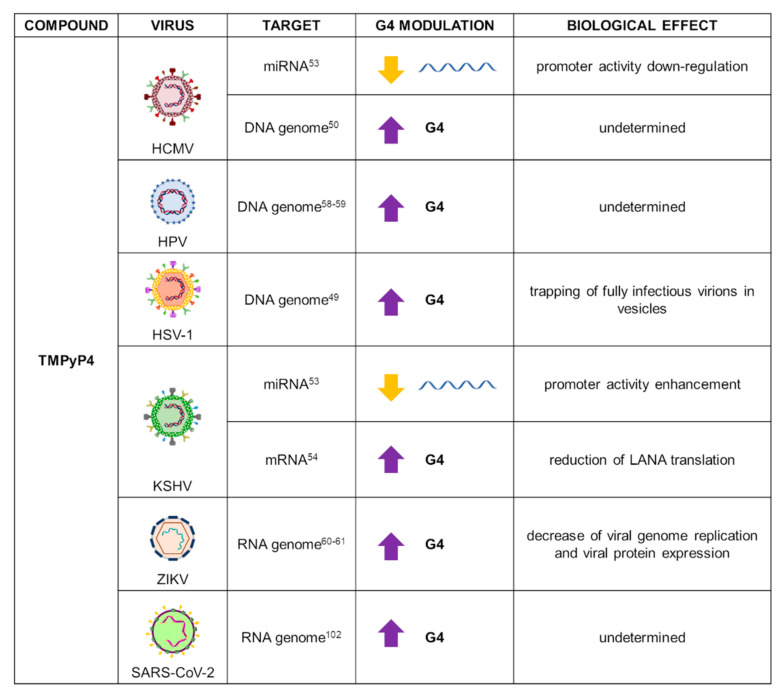
Summary of TMPyP4 applications as a G4 ligand in virology. For each target virus, the site of action (also known as the G4 location), the modulatory effect on G4s, and the overall biological outcome are reported.

**Figure 4 ijms-22-10984-f004:**
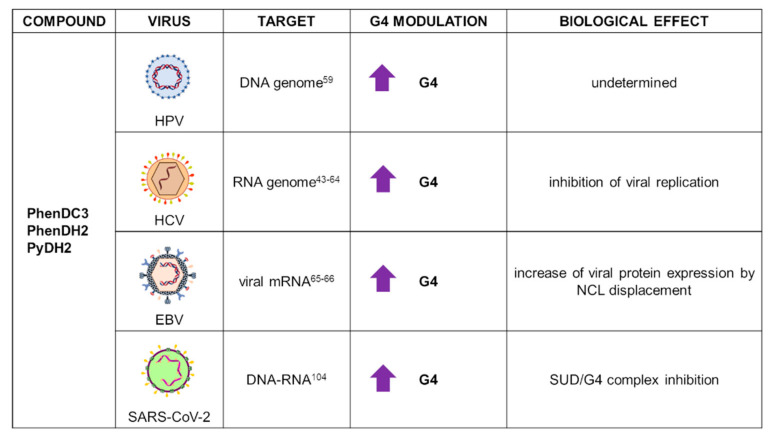
Summary of PhenDC3 and derivative applications as G4 ligands in virology. For each target virus, the site of action (also known as the G4 location), the modulatory effect on G4s, and the overall biological outcome are reported.

**Figure 5 ijms-22-10984-f005:**
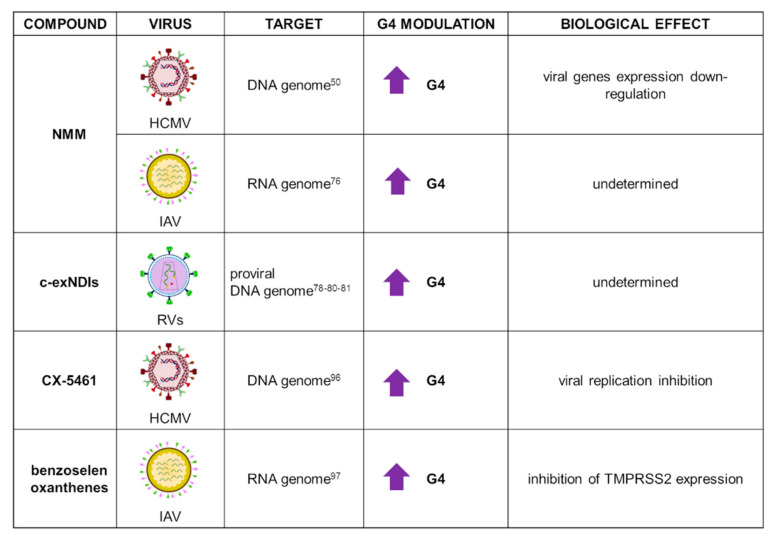
Summary of NMM, c-exNDI, CX-5461, and benzoselenoxanthene applications as G4 ligands in virology. For each target virus, the site of action (also known as the G4 location), the modulatory effect on G4s, and the overall biological outcome are reported.

**Figure 6 ijms-22-10984-f006:**
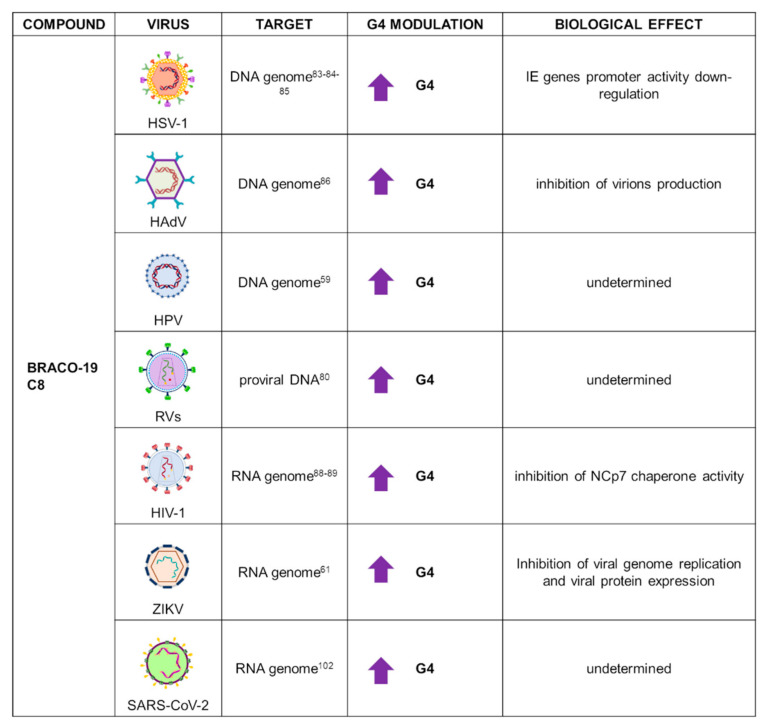
Summary of BRACO-19 applications as a G4 ligand in virology. For each target virus, the site of action (also known as the G4 location), the modulatory effect on G4s, and the overall biological outcome are reported.

**Figure 7 ijms-22-10984-f007:**
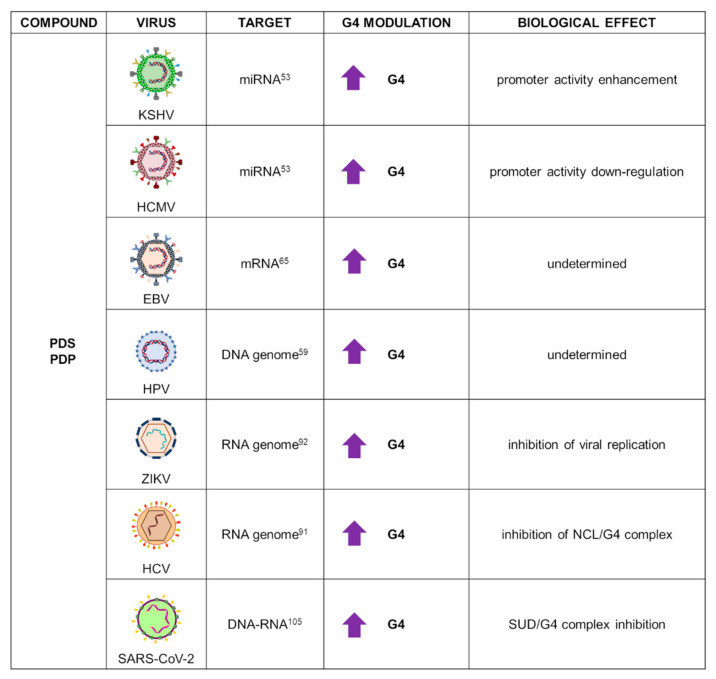
Summary of PDS and derivative applications as G4 ligands in virology. For each target virus, the site of action (also known as the G4 location), the modulatory effect on G4s, and the overall biological outcome are reported.

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
