# Peer review of "G-Quadruplex Targeting in the Fight against Viruses: An Update"

_ijms, 2021, doi:10.3390/ijms222010984_

Round 1
Reviewer 1 Report
Dear Authors,
I appreciated the review and its organization, especially the tables. The issue is important, new antiviral targets. The objectives of your review are clearly presented, and I found that all except one were fulfill: the information about potential G-quadruplexes targets in SARS-CoV-2 is scarce. Example: there are several potential G4s target sequences for G4 ligands, in the SARS-CoV-2 genome, and you show none. I recommend that you should give more information about this matter, and I dare to give you three reading suggestions: 10.1093/bib/bbaa114; 10.1371/journal.pone.0250654; 10.1093/nar/gkab571.
Best regards.
Author Response
We thank the referee for the positive evaluation of our review. We have expanded the SARS-CoV-2 paragraph by including the location of the identified PQSs and their possible implication in SARS-CoV-2 biology. The suggested papers are reported in references 100, 101 and 105.
Reviewer 2 Report
Overall, the manuscript is clear and interesting. The Authors discussed a general overview of G-quadruplex (G4) ligands employed in the fight against viruses (update). They have collected a data on G4 ligands in antiviral therapy from the most recent literature available (including the latest advances against SARS-CoV-2 virus). All Figures showing the summary of G4 ligands applications in virology are very valuable parts of this manuscript. They provide important information regarding the G4 ligands, their target and mode of action. This review is generally well-written and structured, however, it can be accepted if the following minor comments have been addressed:
- Figure 1 presents (A) Chemical structure of G-tetrad and G-quadruplex assembly. (B) virus structure and main components. (C) Baltimore classification of viruses. According to my knowledge and literature, in Baltimore classification in group VII it is a mistake, it should be dsDNA-RT instead of ssDNA.
- In Introduction in paragraph 3 the Authors should correct the name of the HCV abbreviation, i.e. hepatitis C virus instead of hepatitis C.
- In Section 2 named G4 Ligands please clarify sentence: "… great effort has been made to improve binding and selectivity…" of what ?, (…).
- The resolution of Figure 2 is poor, it is difficult to read. The Authors should try to change the resolution of chemical structure of G4 ligands. Moreover, in chemical structures of TMPyP4 and TMPyP2 one hydrogen bond is missing (see schematic illustration in the attachment).
- The Authors should use the same notation for specific phrases throughout the manuscript, e.g. in vitro or in vivo. These phrases were written with "-" symbol in some sentences, while in others without this symbol.
- In Subsection 2.5 BRACO-19 and acridine derivatives in paragraph 4 the abbreviation HAdV refers to singular, while HAdVs to plural.

Author Response
- Figure 1 presents (A) Chemical structure of G-tetrad and G-quadruplex assembly. (B) virus structure and main components. (C) Baltimore classification of viruses. According to my knowledge and literature, in Baltimore classification in group VII it is a mistake, it should be dsDNA-RT instead of ssDNA.
We thank the referee for pointing out this mistake. We corrected as indicated.
- In Introduction in paragraph 3 the Authors should correct the name of the HCV abbreviation, i.e. hepatitis C virus instead of hepatitis C.
We corrected the HCV full name as indicated.
- In Section 2 named G4 Ligands please clarify sentence: "… great effort has been made to improve binding and selectivity…" of what ?, (…).
We thank the referee for noticing this issue. We corrected it as follows: “to improve binding and selectivity of specific ligands towards G4s”.
- The resolution of Figure 2 is poor, it is difficult to read. The Authors should try to change the resolution of chemical structure of G4 ligands. Moreover, in chemical structures of TMPyP4 and TMPyP2 one hydrogen bond is missing (see schematic illustration in the attachment).
We added the missing bond in the porphyrin structures and improved figure resolution.
- The Authors should use the same notation for specific phrases throughout the manuscript, e.g. in vitro or in vivo. These phrases were written with "-" symbol in some sentences, while in others without this symbol.
We corrected the whole text as indicated.
- In Subsection 2.5 BRACO-19 and acridine derivatives in paragraph 4 the abbreviation HAdV refers to singular, while HAdVs to plural.
We thank the referee: we corrected this issue.